# Cannabidiol Effectively Promoted Cell Death in Bladder Cancer and the Improved Intravesical Adhesion Drugs Delivery Strategy Could Be Better Used for Treatment

**DOI:** 10.3390/pharmaceutics13091415

**Published:** 2021-09-07

**Authors:** Shanshan Chen, Changping Deng, Wenyun Zheng, Shihui Li, Yuping Liu, Tong Zhang, Chen Zhang, Yunhui Fu, Hui Miao, Fuzheng Ren, Xingyuan Ma

**Affiliations:** 1Laboratory of Biopharmaceutical and Cell Engineering, School of Biological, East China University of Science and Technology, 130 Meilong Road, P.O. Box No. 365, Shanghai 200237, China; Y30180293@mail.ecust.edu.cn (S.C.); Y10190071@mail.ecust.edu.cn (C.D.); Y30191258@mail.ecust.edu.cn (S.L.); zhang1990chen@126.com (C.Z.); Y12190013@mail.ecust.edu.cn (Y.F.); Y10200042@mail.ecust.edu.cn (H.M.); 2Shanghai Key Laboratory of New Drug Design, School of Pharmacy, East China University of Science and Technology, Shanghai 200237, China; zwy@ecust.edu.cn (W.Z.); Y30191341@mail.ecust.edu.cn (Y.L.); Y53190035@mail.ecust.edu.cn (T.Z.); fzren@ecust.edu.cn (F.R.)

**Keywords:** cannabidiol (CBD), bladder cancer (BC), RNA sequencing, PI3K/Akt, chitosan, CBD-loaded PLGA

## Abstract

Cannabidiol (CBD), a primary bioactive phytocannabinoid extracted from hemp, is reported to possess potent anti-tumorigenic activity in multiple cancers. However, the effects of CBD on bladder cancer (BC) and the underlying molecular mechanisms are rarely reported. Here, several experiments proved that CBD promoted BC cells (T24, 5637, and UM-UC-3) death. For example, T24 cells were treated with 12 µM CBD for 48 h, flow cytometry analysis demonstrated that early and late apoptotic cells were accounted for by 49.91%, indicating CBD enhanced cell apoptosis ability. To deeper explore molecular mechanisms, the CBD-treated T24 cell transcriptome libraries were established. KEGG analysis implied that the significantly changed genes were enriched in the PI3K/Akt pathway. qRT-PCR and Western blot assays verified that CBD regulated BC cells growth and migration and induced apoptosis by inactivating the PI3K/Akt pathway. Meanwhile, the developed chitosan to wrap CBD-loaded PLGA nanoparticles can significantly enhance the adhesion of the material to the mouse bladder wall, and the binding efficiency of mucin to chitosan-PLGA nanoparticles reached 97.04% ± 1.90%. In summary, this work demonstrates that CBD may become a novel reliable anticancer drug and the developed intravesical adhesion system is expected to turn into a potential means of BC chemotherapy drug delivery.

## 1. Introduction

Bladder cancer (BC) is the tenth-leading cause of cancer-related deaths around the world [1], with an estimated 55,000 newly diagnosed cases in 2020 globally [2], and approximately 170,000 deaths [3]. At initial diagnosis, almost 75–80% of cases are non-muscle-invasive bladder cancer (NMIBC) and 20–25% patients develop into muscle-invasive disease during their life [4]. At the present stage, intravesical instillation chemotherapy is the mainstay standard of care for high-risk NMIBC after transurethral resection of the bladder tumor [5]. However, there are still two formidable challenges hindering the therapeutic effect. Chemoresistance gradually occurs in approximately 70% of patients [6]; conversely, interference of bladder physiological characteristics leads to a poor utilization rate of instillation drugs [7]. Therefore, there is an urgent need to find more therapeutic drugs and explore better drug delivery for high-risk NMIBC patients.

In terms of drug discovery, plant-derived products have received considerable attention that possesses characteristics of chemical structural diversity, diverse biologic activities, low adverse reactions incidence, and low costs [8,9]. Notably, it has been extensively studied that diverse anti-cancer compounds from plant sources [10], including vinca alkaloids, taxanes [11], tanshinone [9], and camptothecinsare [12], which are still considered to be the cutting-edge drugs for the treatment of cancers. For example, paclitaxel, extracted from *Taxaceae* family, is a well-known natural anti-cancer drug in the past 30 years. In the clinical treatment of breast cancer, it can cause altered mitosis and cell death by interfering with microtubule function [13,14]. Camptothecin obtained from *Camptotheca acuminata* (*Nyssaceae*) has been approved for treatment of colorectal cancer, ovarian cancer, and small cell lung cancer because of its ability to inhibit DNA topoisomerase [15]. Previously, studies above suggested that many plant-derived compounds had good pharmacological properties. Currently, there is great interest in potential medical use of cannabidiol (CBD) originating from *Cannabis sativa* (also named hemp), a non-psychoactive phytocannabinoid [16]. Studies have shown that CBD can exert multiple beneficial pharmacological effects, including antiepileptic, anxiolytic, anti-inflammatory, and antipsychotic properties [17]. The CBD drug Epidiolex was approved by the Food and Drug Administration (FDA) in 2018 for the treatment of childhood epilepsy, proving the safety and effectiveness of CBD [18]. A large number of studies have focused on its antitumor activity in different types of tumors. For example, Soyeon et al. [19] reported that CBD induced apoptosis in colorectal cancer cells by activating *Noxa* gene expression, and it can promote apoptosis via regulation of *XIAP*/*Smac* in gastric cancer cells [20]. Mohamad and his colleagues indicated that CBD possessed antitumor activity mainly by inhibiting the EGF/EGFR pathway in breast cancer [21]. Although CBD has shown good effects in the treatment of a variety of cancers, reports on treatment of bladder cancer are rarely known.

CBD is a potential chemotherapy drug, but how to increase its utilization in the bladder has become a new challenge. To the best of our knowledge, the residence of free drugs inside the bladder is short due to periodic urination [5]. Given that CBD is highly lipophilic (easy to dissolve in dimethyl sulfoxide (DMSO)) with poor stability (affected by temperature, light, and auto-oxidation [22]), continuous perfusion is required to maintain high drug concentration; thus, this can result in elevated side effects or occurrence of drug resistance. To solve the existing issues, better drug delivery systems are being developed and applied. It has been proven that these challenges can be resolved by encapsulating lipophilic drugs in poly-(lactic-*co*-glycolic acid) (PLGA) particles, thereby allowing extended antitumor activity after a single administration, which is feasible in the long-term treatment of cancer [23]. In addition, chitosan (CS), derived from chitin, has been extensively used as the mucoadhesive materials in the formulation of multiple delivery systems [24]. Furthermore, CS can perform the adhesion function for a long time by reacting with the glycosaminoglycan layer on the surface of the bladder wall [7]. Therefore, coating CS on the surface of PLGA particles is expected to become a novel mucosal delivery system for the treatment of bladder cancer.

In the present study, we respectively investigated the antitumor effects and molecular mechanisms induced by CBD in BC cells and developed a drug delivery system based on positively charged CS coated CBD-loaded PLGA particles. This work deeply explored the mechanism of CBD-mediated anti-tumorogenesis and laid the foundation for the future development of bladder perfusion drug delivery strategy.

## 2. Materials and Methods

Reagents and Antibodies. CBD was purchased from Push (Chengdu, China). The initial concentration of CBD powder dissolved in absolute DMSO was 51 mM and stored at −20 °C in the dark for later use. When using CBD for follow-up experiments, the DMSO concentration needed to be kept below 1‰. Poly-(lactide-*co*-glycolic acid-resomer) (PLGA, Mw: 24,000–38,000 Da, i.v. 0.25–0.35 dL/g) was obtained from Daigang (Jinan, China). Polyvinyl alcohol 1788 (PVA, Mw = 44.05) was purchased from Aladdin (Aladdin, Shanghai, China). RPMI-1640, DMEM, and DMEM/F12 medium were obtained from Gibco (Thermofisher, Shanghai, China). Fetal bovine serum (FBS) was supplied by BI (Israel, South America). Antibiotic-antimycotic solution and 3-(4,5-dimethyl-2-thiazolyl)-2,5-diphenyl-2H-tetrazolium bromide (MTT) were obtained from Solarbio (Shanghai, China). In situ cell death detection kit and RIPA buffer were purchased from Beyotime (Shanghai, China). Antibodies against PI3K (1500 μg/mL, 1:10000), Akt (2504 μg/mL, 1:3000), phospho-Akt (1683 μg/mL, 1:6000), mTOR (500 μg/mL, 1:10000), phospho-mTOR (1000 μg/mL, 1:1000), Bcl-2 (1000 μg/mL, 1:2000), Bax (1600 μg/mL, 1:4000), Cytochrome-c (1000 μg/mL, 1:4000), Caspase 7 (700 μg/mL, 1:1000), Erk1/2 (1000 μg/mL, 1:6000), phospho-Erk1/2 (370 μg/mL, 1:3000), MMP9 (400 μg/mL, 1:600), MMP2 (1000 μg/mL, 1:600), and β-actin (427 μg/mL, 1:5000) were purchased from Proteintech (Shanghai, China). Phospho-PI3K (690 μg/mL, 1:1000) was purchased from Abcolonal (Shanghai, China). HRP-conjugated secondary antibodies were purchased from Proteintech. HPLC-grade methanol and acetonitrile, were obtained from Macklin (Shanghai, China).

Cell Culture. Human T24, 5637, UM-UC-3, and SV-HUC-1 cell lines were supported by Sun Yat-Sen memorial hospital (Sun Yat-Sen University), and their biological characteristics are presented in Appendix A. T24 cells and 5637 cells were cultured in RPMI-1640 medium. UM-UC-3 cells was cultured in DMEM medium, and SV-HUC-1 cells was cultured in DMEM/F12 medium. All media were supplemented with 10% FBS, and 1% antibiotic-antimycotic solution. Cells were cultured at 37 °C under a humidified atmosphere of 5% CO_2_.

Cell Viability Assay. The effect of CBD on the viability of T24, UM-UC-3, 5637, and SV-HUC-1 cells were evaluated quantitatively by the MTT assay. First, 100 µL of the cell suspension was seeded at a density of 2 × 10^4^ cells per well in 96-well culture plated and incubated overnight. Subsequently, the medium was changed, each sample well was added with 200 µL volume of different concentrations of CBD (0 and 1‰ DMSO, 10, 15, 20, 25, 30, 40, 50 µM) at 37 °C in CO_2_ incubator for 48 h. Then, MTT reagent (20 µL of 5 mg/mL) was added into each well and incubated at 37 °C for an additional 4 h. Afterward, the supernatant was carefully removed, and the formed formazan crystals were dissolved by adding 150 µL DMSO to each well. The plates were shaken for 10 min at room temperature; absorbance at 490 nm was measured by using the automatic microplate reader (BioTek, Winooski, Vermont, USA).

Colony Formation Assay. One thousand cells were seeded in 6-well plates in complete medium. The next day the media were changed to fresh medium containing CBD, and after culturing for 48 h, the fresh medium was replaced, and changed every 3 days. After 2 weeks, the medium was discarded, and the cells were washed twice with PBS. The cells were fixed with 4% paraformaldehyde for 15 min, and then the fixing solution was discarded. The cells were then stained with 1‰ crystal violet for 10 min and rinsed with running water slowly. The plates were then observed for the formation of colonies.

Wound-healing Assay. Cell migration was assessed in bladder cancer cells using a wound healing assay. When the density reached 100% confluence, the wound was scratched with a sterile 200 μL pipette tip in the confluent monolayer at the center of culture plates, followed by three washes with PBS. Cells were incubated in medium supplemented with 0.1% (*v/v*) FBS in the presence or absence of CBD or vehicle. Images of the scratches were captured at 0, 6, 12 and 24 h to visually assess cell migration distance. The migratory distance was detected using Image J software (V1.8.0.112, National Institutes of Health, Bethesda, USA).

Hoechst 33258 Staining Assay. The nuclear morphological changes in CBD-treated bladder cancer cells were evaluated using the Hoechst 33258 stain (Solarbio, Shanghai, China). Briefly, equal number of cells was seeded in 6-well plates overnight. Cells were then washed twice with PBS and stained for 10 min with 10 µg/mL Hoechst 33258 at 37 °C in the dark after treating the cells with CBD or vehicle for 48 h. Then dyeing medium was removed and the wells were washed twice more with PBS, and the nuclear morphology of the cells was observed under a fluorescence microscope (Olympus, Tokyo, Japan).

Terminal Deoxynucleotidyl Transferase (TdT) dUTP Nick-End Labeling (TUNEL) Assay. TUNEL assay was used to detect apoptosis and was performed using in situ Cell Death Detection Kit (Beyotime, Shanghai, China). Bladder cancer cells were seeded in confocal dishes at 37 °C. After 24 h of culture, culture medium containing 12 µM of CBD was added and further cultured for 48 h in the incubator. Cells were washed twice with PBS and fixed with 4% paraformaldehyde for 30 min. Cells were then permeabilized with PBS containing 0.3% Triton X-100 at room temperature for 5 min. After washing the cells three times with PBS, TUNEL reaction mixture from the assay kit was used to co-incubate with cells for 1 h at 37 °C in a 5% CO_2_ incubator. The dishes were washed with PBS and stained with 10 µg/mL Hoechst 33258 for 10 min at 37 °C. These cells were observed by confocal microscopy (Olympus, Tokyo, Japan).

Apoptosis Assay. The Annexin V-FITC/Propidium Iodide (PI) (Elabscience, Wuhan, China) double staining assay was performed according to the manufacturer’s manual. Briefly, bladder cancer cells were seeded at a density of 5 × 10^5^/well in 6-well plates overnight and then stimulated by 1‰ DMSO or 12 µM CBD as the vehicle control or experimental group. After treating for 48 h, the cells and culture medium were harvested using trypsin-EDTA and centrifuged at 500 rpm for 5 min at room temperature. The supernatant was discarded, and pellet was resuspended in PBS. Cell suspension was centrifuged under the same condition. The supernatant was discarded and resuspended by adding 100 µL of 1× binding buffer. Then, 2.5 µL of Annexin V-FITC and PI staining solution were added to untreated and treated cell suspension, mixed well, and incubated for 15 min at room temperature without light. Finally, 100 µL of 1× binding buffer was added to the suspension again. Fluorescence intensity was measured using the Beckman flow cytometer (Beckman Coulter, Inc.250 S.Kraemer Boulevard Brea, CA 92821, USA), and the apoptotic rates of CBD-treated cells were analyzed by using FlowJo software (FlowJoV10, Becton, Dickinson & Company, New York, NY, USA).

RNA Sequencing (RNA-seq) Assays. The extracting RNA from CBD-treated cells (Appendix A) were used to RNA-seq. The transcriptome libraries were constructed by Shanghai Majorbio Technology Co., Ltd (Shanghai, China). All quality inspection data were presented in Appendix A. The mRNA expression data was uploaded to the DEseq2 software website (https://login.majorbio.com/login, accessed on 7 September 2021). To identify differential expression genes (DEGs) between two different samples, the expression level of each transcript was calculated according to the transcripts per million reads (TPM) method. In addition, Kyoto Encyclopedia of Genes and Genomes (KEGG) functional-enrichment analysis was performed to identify which DEGs were significantly enriched in metabolic pathways at Bonferroni-corrected *p* value ≤ 0.05 compared with the whole-transcriptome background. KEGG pathway analysis was conducted by Goatools (https://github.com/tanghaibao/Goatools, accessed on 7 September 2021) and KOBAS (http://kobas.cbi.pku.edu.cn/home.do, accessed on 7 September 2021).

Quantitative Real-time PCR Analysis. Total RNAs extraction kit, the First-Strand cDNA synthesis kit, and qRT-PCR kit were purchased from Promega (Shanghai, China). All the indicated samples were normalized to actin and then relative mRNA levels were calculated by using the comparative Ct method. The qRT-PCR primers (Appendix A) were synthesized by Sangon (Shanghai, China).

Western Blotting Analysis. Cells were lysed by Radio Immunoprecipitation Assay (RIPA) buffer with 1 mM PMSF to obtain the proteins for Western blotting. The BCA Protein Assay Kit (Sangon, Shanghai, China) was used to quantify the protein concentration. Subsequently, Western blotting experiments were performed in accordance with the previously reported laboratory procedures [25].

Preparation of CS-coated CBD-loaded PLGA Nanoparticles. Bernhard Brauner et al. [26] had been confirmed that PLGA nanoparticles (PLGA NPs) are more suitable than PLGA microparticles for instillative therapy. Hence, following a solvent evaporation technique, PLGA NPs loaded with CBD were prepared from an oil-in-water emulsion. Briefly, PLGA (25 mg) and CBD were dissolved in 2 mL of mixed reagent (V_acetone_:V_absolute ethanol_ = 4:1). The oil phase was added dropwise to 20 mL of a 1% (*w/v*) PVA aqueous solution under stirring at 800 rpm in a magnetic stirrer for 5 h to allow solvent evaporation. Then, the NPs were collected by centrifuge and washed with 10 mL of ultrapure water three times in order to eliminate remnants of PVA, namely CBD-loaded PLGA NPs (CBD/PLGA NPs). Next, samples were resuspended in 10 mL ultrapure water and subsequently poured into equal volume of 0.16% (*w/v*) CS solution and stirred at 800 rpm in a magnetic stirrer for 2 h at room temperature. Finally, 500 µL ultrapure water was added as a cryoprotectant, and samples were frozen at −80 °C overnight and freeze-dried for 24 h using a freeze dryer, namely CS-coated CBD-loaded PLGA NPs (CS-CBD/PLGA NPs).

For the internalization and adhesion experiments, DiI (Beyotime, Shanghai, China) was used as the fluorescent agent, DiI-loaded PLGA NPs (DiI/PLGA NPs) and CS-coated DiI-loaded PLGA NPs (CS-DiI/PLGA NPs) were prepared using the aforementioned protocol.

Characterization of CS-CBD/PLGA NPs. The morphologies of NPs (Blank PLGA NPs, CBD/PLGA NPs and CS-CBD/PLGA NPs) were observed using transmission electron microscope (TEM) (JEM-1400, JEOL, Tokyo, Japan). For TEM examination, the lyophilized NPs were dissolved and placed onto formvar-coated copper grid and negatively stained with phosphotungstic acid. The size distribution, zeta potential, and PDI of NPs were measured by dynamic light scattering technique (DLS) (Malvern, Malvern city, UK). Each sample was measured in triplicate.

Determination of EE. The lyophilized NPs (CBD/PLGA NPs and CS-CBD/PLGA NPs) were dissolved in 1 mL DMSO for 6 h and then were centrifuged for 30 min at 4 °C at 14,000 rpm. The supernatant was collected and determined by high performance liquid chromatography (HPLC) (Agilent 1200 series, Agilent Technologies Inc., CA, USA ) equipped with a reverse-phase C18 (250 × 4.6 mm, 5 µm) column. The mobile phase consisting of acetonitrile (A) and water (B) was used for separation of CBD through isocratic elution at a flow rate of 1 mL/min and an injection volume of 10 µL. CBD detection and quantification wavelength was set at 220 nm. The EE was calculated using the following formula.


$$\mathrm{EE}\left(\%\right)=\frac{\mathrm{Detected}\;\mathrm{amount}\;\mathrm{of}\;\mathrm{CBD}\;\mathrm{in}\;\mathrm{the}\;\mathrm{NPs}\;}{Initial\;amount\;of\;CBD\;in\;the\;NPs}\times 100\%$$


Determination of Fourier Transform Infrared (FTIR) Spectroscopy. FTIR spectra of samples homogeneously mixed with KBr and compressed into discs were traced in the range of 400 to 4000 cm^−1^, using FTIR spectrophotometer (Shimadzu, Kyoto, Japan). 

Stability Studies. The different formulations were stored at 4 and 25 °C for a period of 5 weeks. Samples were taken after 0, 2, and 5 weeks and the following physicochemical properties were examined, including particle size and encapsulation efficiency (EE,%).

In Vitro Assay of Drug Release Studies. Briefly, 10 mg of NPs were dispersed in 1 mL of phosphate buffered saline (PBS containing 1% (*v/v*) Tween 80) with different pH (i. e., 5.0 and 6.5) and then incubated at 37 °C at 100 rpm. At predetermined time intervals, microcentrifuge tubes containing CBD-NPs in PBS were centrifuged at 12,000 rpm for 10 min at 4 °C, and then collected 100 µL clear solution of supernatants to analyze by HPLC. The release medium was replaced by fresh dissolution medium at each interval; 100 µL samples were then withdrawn and replenished with fresh media.

In Vitro Assay of Mucoadhesion Studies. The adsorption–association of mucin with the particles was used as a method to assess mucoadhesive properties of the particles prepared [27]. Here, 2 mL of mucin suspension (0.5 mg/mL) and 2 mL of 2 mg/mL NPs dispersion (Blank PLGA NPs and CS-PLGA NPs) was mixed (vortexed) and incubated in a shaker at 37 °C at 100 rpm for 2 h. The mixtures were then centrifuged at 14,000 rpm for 30 min, and the supernatant was collected and used for the measurement of free mucin.

Bradford reagent is an effective means to detect the concentration of free mucin to further evaluate the amount of mucin adsorbed on the NPs. According to the Bradford standard curve, 10 µL of all samples (of known and unknown mucin concentration) are mixed with 190 µL Bradford reagent, and then used to detect the absorbance at a wavelength of 595 nm. Finally, the mucin content of the samples was calculated and determined.

In Vivo Assay of Mucoadhesion Studies. Mucoadhesion test was performed according to the previously described method by Martin et al. [28]. Mice were anaesthetized by intraperitoneal injection of phenobarbital sodium. External genitalia were cleansed and urine in their bladders were emptied. The bladders were then filled with PBS with or without CS-modified, DiI-loaded NPs (2 mg/mL DiI/PLGA NPs and CS-DiI/PLGA NPs) for 2 h. Mice were sacrificed, and bladders were washed extensively with PBS to remove non-adherent NPs. Samples were frozen, sliced, and captured with CLSM.

Cellular Uptake Analysis. T24 cells with density of 5 × 10^5^ cells/well were grown on confocal dishes overnight. A volume of 100 µL NPs including DiI/PLGA NPs and CS-DiI/PLGA NPs was incubated for 2 h and 6 h at 37 °C. Then, the excess NPs were washed by fresh PBS for three times. The cells were fixed by 4% paraformaldehyde for 15 min at room temperature, washed three times with PBS, and then incubated with 10 µM Hoechst 33258 dye for 5 min. After three washes with PBS, cells were captured by confocal laser scanning microscope (CLSM, Nikon, Shanghai, China).

Statistical Analysis. All data are presented as the mean ± standard deviation (SD) of the mean following analysis with GraphPad Prism 5.0 (GraphPad Software, San Diego, CA, USA). Statistical significance values were evaluated through one-way ANOVA test with post hoc contrasts by Student–Newman–Keuls test, or part of the data were conducted by Student’s t test, using SPSS 22.0 software for evaluation. *p* < 0.05 was defined as statistically significant difference.

## 3. Results and Discussion

### 3.1. CBD Inhibited Cell Viability and Induced Apoptosis in BC Cells

In order to evaluate the effect of CBD (Figure 1A) on BC in vitro, the MTT assay was performed by using different CBD concentrations for 48 h in various bladder cells, including cancer cells T24, UM-UC-3, 5637, and normal bladder cells (NBC) SV-HUC-1. As shown in Figure 1B, CBD (0–25 µM) inhibited the viability of bladder cells in a concentration-dependent manner, except for the NBC SV-HUC-1. The half maximal inhibitory concentration values (IC_50_) were calculated: 10.85 ± 2.18 µM, 21.83 ± 1.32 µM, 22.92 ± 0.97 µM, 40.68 ± 1.87 µM for T24, UM-UC-3, 5637, and SV-HUC-1 cells, respectively. As a result, T24 cells were more sensitive to CBD compared with other cell lines. To examine the effect of CBD treatment on clonogenic survival, the cell colony formation assay was performed. This ability was reduced by CBD treatment, especially for T24 cells (Figure 1C,D).

To assess whether the decreased viability of CBD-induced cells could be related to apoptosis, Hoechst 33258 nucleus staining was performed. Compared to the uniformly distributed blue fluorescence in the control group, CBD-treated BC cells for 48 h increased the proportion of apoptotic cells characterized with chromatin condensation and cell nuclei shrinkage (Figure 1E). In addition, it was also clearly observed that T24 cells treatment with 12 μM CBD resulted in a decrease in cell confluence due to the CBD concentration reaching the IC_50_ value. Subsequently, cell apoptosis was confirmed by TUNEL staining and confocal laser scanning microscope (CLSM). The results (Figure 1F) revealed that CBD-treated for 48 h could significantly increase TUNEL-positive cells compared with no CBD, especially T24 cells. Compared to the UM-UC-3 and 5637 test groups, T24 cells exited stronger green fluorescence intensity and chromatin condensation under CBD treatment. Flow cytometry was utilized to further quantitative detect cell apoptosis after CBD treatment using the Annexin V-FITC/PI double staining. Figure 1G shows that exposure of T24 cells to 12 µM CBD for 48 h elevated the apoptotic cell population (early and late apoptotic cells, accounting for 49.91%) compared with the controls, but basically had no proapoptotic effect on UM-UC-3 and 5637 cells (nearly 4.35% and 8.60%). In general, CBD can cause T24 cells apoptosis at lower concentrations (12 µM) compared with other BC cells.

Here, the results above confirmed that CBD inhibited the growth of BC cells, and the inhibitory effect was concentration dependent. However, their IC_50_ values were significantly different, with the highest (approximately 23 µM) in 5637 and the lowest (approximately 12 µM) in T24 cell line. To explore how CBD could promote BC cells death at low concentrations, 12 µM was selected as the suitable concentration for the further studies. Moreover, the colony formation results demonstrated that CBD inhibited BC cells proliferation effectively, which were consistent with the MTT assay. CBD-induced apoptosis of the present study was in agreement with previous published studies, which showed that CBD could induce apoptosis in various cancers, such as colorectal cancer [19], gastric cancers [20] or breast cancer cell lines [21]. CBD-treated T24 (12 µM) may cause a decreased number of living cells, and an increased number of TUNEL-positive and Annexin V/PI double-positive cells. While under the same concentration, almost no cell apoptosis was detected in UM-UC-3 and 5637 cells. These results suggest that UM-UC-3 and 5637 could be more resistant to CBD than T24, while T24 cells were more sensitive to CBD.

### 3.2. CBD Inhibited BC Cells Migration

The effect of CBD on the migration of BC cells was examined by wound healing assay. After scraping for 6, 12, and 24 h, the migration of BC cells to the cell-free area was measured in the absence of CBD and 12 µM CBD treatment. CBD treatment markedly inhibited these cell lines migration, except for T24 cells at 6 h (Figure 2A). The scratched wound was almost completely closed in the control group at 24 h. Figure 2B represents the mean values of cell migrated area at 6, 12, and 24 h. At the end of 24 h, the percentage of acellular area in T24, UM-UC-3 and 5637 was 23.08%, 12.96%, and 7.98% in the absence of CBD, whereas treated with CBD 12 µM had an acellular area of 47.16%, 58.68%, and 27.81%, respectively. According to the results of MTT, 12 µM CBD had no significant effect on the proliferation of UM-UC-3 and 5637 cells. However, the scratch test found that 12 µM CBD could effectively inhibit the migration of UM-UC-3 and 5637, which was better than T24. Although it was temporarily unable to solve this phenomenon, the existing results indicated that CBD could inhibit the migration of BC cells in a time-dependent manner. This results were consistent with previous reports that CBD strongly reduced migratory ability, such as endometrial cancer [29], glioma [30], and polymorphonuclear leukocyte [31]. In summary, these findings implied that CBD could possess great potential application in inhibiting cell proliferation, migration, and inducing apoptosis in BC cells.

### 3.3. Global Transcriptomic Analysis of CBD-Treated T24 Cells

The transcriptomic profile of T24 cells was constructed by RNA sequencing (RNA-seq), there were four transcriptome libraries, including two control groups (no CBD treatment) and two test groups (CBD treatment for 48 h). As a result, a total of 200,500,104 raw reads were generated, and 199,478,052 clean reads (99.50%) were obtained. The mean Q20, Q30, and GC content were 96.83%, 91.79% and 50.85%, respectively (Appendix A). For biologically repeated experiments, DESeq2 already conducted experimental error control; thus, the standard for screening differential genes is generally: |log_2_FC| > 1 and *p* adjust < 0.05. The differential expression gene (DEGs) are shown as volcano plots in Figure 3A. The results show that 1711 and 764 DEGs were significantly upregulated and downregulated in CBD-treated T24 cells compared with the control group. Next, Kyoto Encyclopedia of Genes and Genomes (KEGG) enrichment analysis was performed. The pathway reports of all DEGs enrichment are shown in Figure 3B. Results showed that human papillomavirus infection (70), PI3K/Akt (68), and cytokine-cytokine receptor interaction (53) signaling pathways were significantly enriched in CBD-induced T24 cells. Multiple studies have delineated that PI3K/Akt was a key signaling involved in different cancer cells behaviors, especially cell proliferation, apoptosis, and migration [26]. Hence, based on existing reports, we speculated that PI3K/Akt signaling pathway could be involved in mediating CBD on the biological activity of BC.

### 3.4. CBD Inhibited Cell Proliferation, Migration, and Enhanced Apoptosis by Regulating PI3K/Akt Pathway-Related Genes Expression

By analysis of the CBD-treated T24 cells transcriptome database, it was found that the related genes expression level of PI3K/Akt downstream were changed after 48 h compared with the control group (Figure 4A). Among them, *Bcl-2* and *MMP9* were downregulated by 0.51- and 0.09-fold, while *Bax*, *Caspase 7*, and *Cyto-c* were upregulated by 1.47-, 1.01-, and 2.27-fold. Based on the results above, these genes expression in CBD-treated T24 cells were confirmed by qRT-PCR and Western blotting. As shown in Figure 4B, the mRNA expression levels of *Bcl-2*, *MMP9* were decreased compared with the control, but *Bax*, *Cyto-c*, and *Caspase 7* were increased, respectively. The *MMP2* expression level was not markedly changed, and the Western blotting analysis displayed that the expression of these genes at protein levels. As shown in Figure 4C–E, the results show that the expression of p-PI3K, p-Akt, p-mTOR, p-Erk1/2, Bcl-2, MMP9 in CBD-treated T24 cells significantly decreased, while the Bax, Cyto-c, Caspase 7 markedly increased in a time-dependent manner. There was no appreciable effect on PI3K, Akt, mTOR, Erk1/2, and MMP2 in CBD-treated T24 cells (Figure 4F). The reason for this result is because several of these genes are required to be phosphorylation to function [19]. Therefore, there was no change in the protein level when it was not phosphorylated.

It has been reported that the phosphatidylinositol 3-kinase (PI3K)/Akt pathway is involved in a variety of biological processes and is often abnormally activated in various cancers [32]. Increasing evidence shows that cellular processes such as cell proliferation, apoptosis, and migration are directly or indirectly regulated by PI3K/Akt signaling [33]. In addition, emerging evidence further confirmed that the PI3K/Akt signaling pathway can affect different genes and cause multiple biological functions [34]. For instance, the activation of mTOR by Akt played an important role in cell proliferation, survival, and growth [35]. From the studies above, the present results concluded that CBD can inhibit T24 cells proliferation via inactivation of PI3K/Akt/mTOR pathway. Moreover, it has been already proven that Bcl-2 family is a significant downstream gene of Akt in the PI3K/Akt signaling pathway and is involved in mediating apoptosis [36], which includes pro-apoptotic proteins and anti-apoptotic proteins. Bax, as the member of pro-apoptotic protein, promotes cell apoptosis by activating *Caspase* expression and releasing Cytochrome c (Cyto-c) from mitochondria, while Bcl-2, an anti-apoptotic Bcl-2 family protein, restrains cell apoptosis by blocking the release of Cyto-c. Cyto-c owns the ability of activating *Caspase* 9, and then *Caspase* 9, in turn, activates *Caspase 7*, thus triggering apoptosis [37]. To further confirm whether the mechanism of CBD-treated T24 cells was consistent with the previous reports in other cells, the expression levels of *Bcl-2*, *Bax*, *Cyto-c*, and *Caspase 7* in CBD-treated T24 cells were examined at the transcription and protein levels. The results showed that CBD could dramatically downregulate the expression of Bcl-2, but upregulate the expression of Bax, Cyto-c, and Caspase-7 in T24 cells. Taken together, these results confirmed that the PI3K/Akt/Bcl-2 pathway was involved in the CBD-induced apoptosis of T24 cells.

In terms of tumor invasion, the degradation of the extracellular matrix (ECM) plays a considerable role [38]. In addition, ECM is mainly degraded by proteolytic enzymes, matrix metalloproteinases (MMPs) family, and as the most important, are widely involved in cell migration in vitro, such as MMP2 and MMP9 [39]. Furthermore, several studies revealed that MMPs expression is affected by the crosstalk between the PI3K/Akt and the Erk1/2 signaling pathway [40,41]. In this study, the results revealed that CBD significantly suppresses the expression levels of p-Akt, p-Erk1/2, and MMP9 in T24 cells, which is consistent with previous studies [42]. Briefly, these findings suggest that CBD mainly inactivates PI3K/Akt and Erk1/2 signals, thereby downregulating MMP9 gelatinolytic activity and ultimately affecting migration. Therefore, CBD plays an important anti-tumor activity in BC cells mainly by preventing the activation of PI3K/Akt and Erk1/2/MMP9 pathways. Although CBD has a good pharmacological effect on BC cells, it is difficult to exert all the pharmacological functions due to its poor water solubility. Thus, the other goal of this work is to develop a drug delivery system that achieves intravesical adhesion.

### 3.5. Characterization of CBD-Loaded and CS-Coated PLGA NPs

The representative transmission electron microscope (TEM) images of blank PLGA NPs, uncoated (CBD/PLGA NPs), and coated NPs (CS-CBD/PLGA NPs) are shown in Figure 5A,B and Appendix A, respectively. The CBD/PLGA NPs and CS-CBD/PLGA NPs all presented smooth surfaces and spherical morphology. The typical particle size, PDI, and zeta potential determined by dynamic light scattering (DLS) of nanoparticles (NPs) is listed in Appendix A. The size and PDI of CS-CBD/PLGA NPs were approximately 287.20 ± 0.90 nm and 0.134 ± 0.025, which was larger than that of CBD/PLGA NPs (192.90 ± 2.41 nm and 0.041 ± 0.027). Compared with CBD/PLGA NPs (−6.270 ± 0.927 mV), the zeta potential of CS-CBD/PLGA NPs was 3.370 ± 0.158 mV. Studies had confirmed that the protonated amino group of CS could react with the carboxyl group of PLGA to form intermolecular hydrogen bonds, such that CS could be adsorbed on the surface of PLGA [43]. Furthermore, the encapsulation efficiency (EE) of CBD-loaded bare and CS-coated PLGA NPs were examined to be 70.31 ± 0.69% and 78.52 ± 0.82%, respectively (Figure 5C). Next, Fourier transform infrared spectroscopy (FTIR) was used to detect whether the components were synthesized correctly. The infrared spectra of PLGA NPs, CS, and CS-PLGA NPs are displayed in Figure 5D. For PLGA NPs, the carboxyl group (C=O) of PLGA exhibited stretching at 1758.5 cm^−1^. CS alone showed a characteristic peak at 1655.2 and 3447.6 cm^−1^ (C-O stretching, O-H stretching, and N-H stretching). The spectrum of CS-PLGA NPs showed that the characteristic absorption peaks of PLGA and CS were 1757.8 and 1632.7 cm^−1^, respectively, and a stronger peak representing OH stretching in 3446.4 cm^−1^ was demonstrated. These spectral characteristics are consistent with other reports [27]. As a result, existing data suggest that CS was successful coated in PLGA surface. All these results indicate that the uniform NPs with CS modification had been successfully fabricated. The stability of the CBD/PLGA NPs and CS-CBD/PLGA NPs was determined by evaluating the changes in size and EE over time Figure 5E and Appendix A). The results indicate that the CS-CBD/PLGA NPs was stable for 5 weeks at 4 °C. In addition, release of CBD from bare and CS-coated PLGA NPs in various pHs were characterized by using high performance liquid chromatography (HPLC). Different pHs represented different situations, such as the pH 5.0-represented lysosome, and the pH 6.5 was the same as the tumor microenvironment and urine environment. As shown in Figure 5F, the in vitro release of CBD from PLGA NPs presented a typical biphasic release pattern, including initial burst release until the first 15 h, followed by a continuous slowly release mode, lasting up to 110 h. At pH 6.5, the initial burst rate was 34.64 ± 2.22% within 15 h, and the cumulative release rate reached 49.85 ± 2.76% at 110 h. However, within 15 h in lower pH, the burst release rate was 50.55 ± 2.02%, and at 110 h, the cumulative release rate reached 69.07 ± 2.46%. For CS-CBD/PLGA NPs, all release profiles had a slow drug release, the cumulative release rate was less than 20% (pH 6.5: 13.54 ± 0.40%; pH 5.0: 15.67 ± 0.78%) (Figure 5G). This result may be due to CS being coated on the surface of PLGA NPs will reduce the burst release of an encapsulated drug [44]. Notably, the low release rate can ensure the minimal leakage of the drug during transport in the circulation.

### 3.6. In Vitro and In Vivo Adhesion and Cellular Uptake of NPs

As shown in Figure 6A, the binding efficiency was used to determine the adsorption capacity of mucin to NPs. At 2 h, the binding efficiencies of mucin to CS-PLGA NPs (97.04 ± 1.90%) were significantly greater than that of PLGA NPs (80.73 ± 2.80%) (*p* < 0.05), indicating the CS-modified NPs had excellent adhesion. Figure 6B shows in vivo fluorescence images of a mouse after instillation into the bladder of DiI-loaded bare and CS-coated PLGA NPs. The fluorescence signal intensity of CS-DiI/PLGA NPs in the bladder was 6.54-fold that of DiI/PLGA NPs (Figure 6C), suggesting that CS modification increased bladder instillation and prolonged the retention time. This property enhances the adhesion between the material and the bladder wall, thereby increasing drug absorption and prolonging the medication time.

To verify the T24 cells uptake ability of DiI/PLGA NPs and CS-DiI/PLGA NPs, the red fluorescence intensity of DiI released from these NPs was detected by CLSM. As shown in Figure 6D, it was found that the red fluorescence intensity of CS-DiI/PLGA NPs in the cells after 2 h incubation was higher than that of DiI/PLGA NPs, which indicates that the cells had absorbed the NPs. This phenomenon further confirmed that CS-PLGA NPs can effectively avoid the periodic urination of the bladder every 2 h. When the duration was prolonged to 6 h, CS-DiI/PLGA NPs had a higher cellular uptake (*p* < 0.05). Besides, CS-DiI/PLGA NPs were dispersed in the cytoplasm and nuclear of the T24 cells. Representative histograms illustrated a higher fluorescence intensity for T24 cells treated with CS-DiI/PLGA NPs compared to that of DiI/PLGA NPs control (Figure 6E). For PLGA NPs, it can be taken up by cells through endocytosis, but it is interfered by electrostatic repulsion, ultimately resulting in lower uptake efficiency of PLGA NPs than CS-PLGA NPs.

Studies reported that CS was a natural cationic polysaccharide, and it could interact with the negatively charged mucin (the main components of bladder mucosa) to drive strong mucosal adhesion [45]. Moreover, CS easily fused with negatively charged cell membranes [46]. The results of this study were consistent with Martin et al. [28], it had been successfully confirmed that CS modification could enhance the adhesion and uptake ability of bare PLGA NPs.

### 3.7. In Vitro Cytotoxicity of CBD-Loaded and CS-Coated PLGA NPs

The cytotoxicity of CBD-loaded bare and CS-coated PLGA NPs were evaluated in T24 cells and SV-HUC-1 cells, and the results are shown in Figure 7. The cell viabilities of only CBD dissolved in medium against T24, and SV-HUC-1 cells had no significant toxicity at any tested concentrations, including at 50 µM (Figure 7A,B,E,F). The obtained results confirmed that CBD dissolved in medium alone could not exert its pharmacological activity. However, the cell viabilities of CBD dissolved in DMSO against T24 cells and SV-HUC-1 cells decreased in a concentration- and time-dependent manner. The blank PLGA NPs had no significant cytotoxicity to T24 and SV-HUC-1 cells at a polymer concentration from 10–50 µM (corresponding to the concentration of CBD used in the cytotoxicity) (Figure 7C,D,G,H). This reflected that, including at the highest polymer concentration, the cell viability still exceeded 90%, which indicates the safety of the drug delivery system. In addition, CBD/PLGA NPs and CS-CBD/PLGA NPs could inhibit the proliferation of T24 cells in a time- and concentration-dependent manner, without causing damage to NBC SV-HUC-1. In general, at the in vitro cellular level, the therapeutic effect of CS-CBD/PLGA NPs per unit time was not as good as CBD/PLGA NPs, and worse than free CBD. The reason was that the PLGA was degraded by ester bond hydrolysis, which led to in rapid drugs [7]. CS was coated on the outer surface of PLGA NPs, and it slowed down the release of the drug more [26]. Due to the difficulty of in situ modeling of mouse bladder cancer, animal experiments could not be performed. In the future, in which laboratory technology allows, we also hope to be able to verify the specific role of materials in vivo. Taken together, all these results confirm that CS-coated PLGA NPs provided a safe delivery system, which not only helps CBD dissolved in water and fully exert its anti-tumor pharmacological activity but also does not cause toxic side effects of NBC.

## 4. Conclusions

In conclusion, MTT and colony forming assays showed that CBD could inhibit proliferation on BC cells, including T24, UM-UC-3, and 5637. Especially at 12 µM, the effect on T24 was the most significant. Moreover, cell apoptosis was detected by using nuclear staining, TUNEL, and flow cytometer. Subsequently, wound healing assay suggested that CBD reduced the migration ability of BC cells. In order to explore the potential reasons of CBD to cause the results above, the CBD-treated T24 cells transcriptome analysis was conducted. RNA sequencing results indicated that several genes related to apoptosis, proliferation, and migration underwent significant changes, such as *Bax*, *Cyto-c*, *Bcl-2*, and *Caspase 7*. Further analysis by KEGG found that the significantly changed genes were enriched in the PI3K/Akt signaling pathway. Therefore, these related genes were screened from the transcriptome databases and confirmed their expression levels by qRT-PCR and Western blotting. These results indicate that CBD increased Bax, Cyto-c, and Caspase 7 levels, whereas it decreased Bcl-2, p-PI3K, p-Akt, p-mTOR, p-Erk1/2 and MMP9. Taken together, CBD can inhibit BC cells growth, migration, and induce apoptosis by inactivating of the PI3K/Akt pathway. Meanwhile, using chitosan to wrap CBD-loaded PLGA nanoparticles may significantly enhance the adhesion of CBD in the bladder wall, which may not only avoid damage caused by repeated perfusion of organic solvents, but also achieve the purpose of long-term treatment. We believe that our study makes a significant contribution to the field because these results can be developed as a promising strategy for a safer and more efficient anticancer therapy.

## Figures and Tables

**Figure 1 pharmaceutics-13-01415-f001:**
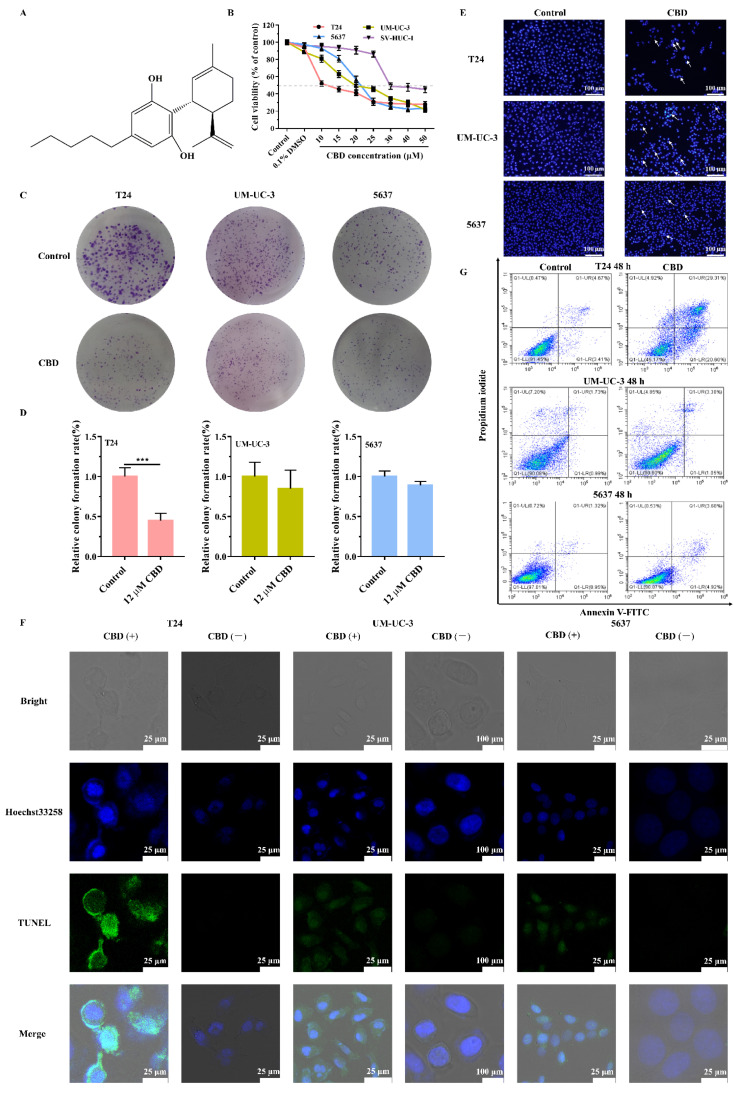
Effect of CBD on BC cells viability and apoptosis. (**A**) Chemical structure of CBD. (**B**) NBC (SV-HUC-1) and various BC cells were treated with 0 and 1‰ DMSO, 10, 15, 20, 25, 30, 40, 50 µM of the CBD for 48 h. Cell proliferation was examined by MTT assay. (**C**) BC cells were treated with 0 or 12 µM of CBD. After 2 weeks, the cells were stained with crystal violet, and the colonies were photographed using a digital camera. (**D**) Statistical analysis of the colony rate of BC cell population after CBD treatment. (**E**) Nuclear fragmentation and condensation (Hoechst 33258 staining) were detected by fluorescence microscopy (magnification, ×100) after treatment with 12 µM CBD for 48 h. All of these signs are representative of apoptosis. (**F**) Cell apoptosis was detected with TUNEL assay, and Hoechst33258 was used as a co-stain to dye the nuclei of the BC cells. After treatment of CBD (12 μΜ), damaged DNA was visualized in bright green (TUNEL-positive cells), indicating apoptosis. (**G**) Cells stained with Annexin V-FITC and PI were studied using flow cytometry to quantitatively detect the apoptosis induced by exposure to 12 μM CBD in BC cells. Values are shown as mean ± SD of three independent experiments. *** *p* < 0.001 compared to the control.

**Figure 2 pharmaceutics-13-01415-f002:**
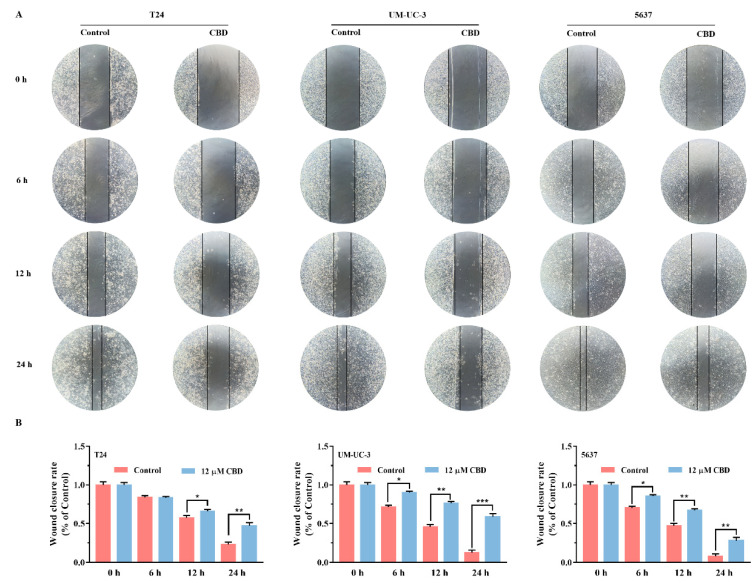
Effect of CBD on cell migration in BC cells. (**A**) Representative images of cell migration of BC cells assessed by the wound healing assay. BC cells were treated with or without 12 μM CBD and photographed at 0, 6, 12, and 24 h after the wound. (**B**) Mean values of cell migration area (represented by % of total image area) were obtained from various conditions in panels (**A**). * *p* < 0.05, ** *p* < 0.01, *** *p* < 0.001 compared to the control.

**Figure 3 pharmaceutics-13-01415-f003:**
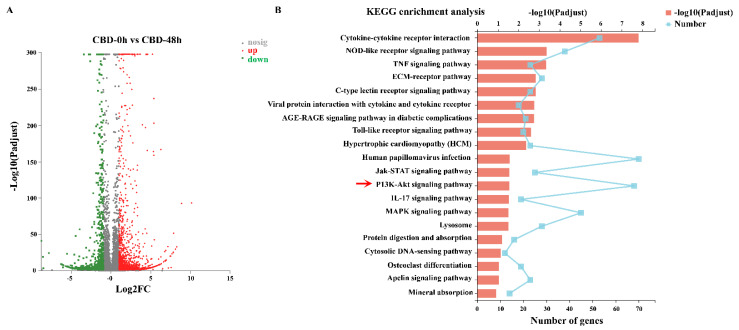
RNA sequencing of T24 cells after CBD exposure for 48 h. (**A**) T24 cells were treated with 12 µM CBD for 48 h and RNA-seq was performed to investigate DEGs. The number of up- and downregulated genes in response to CBD in T24 cells is indicated. (**B**) KEGG pathway enrichment analysis in CBD-exposure group.

**Figure 4 pharmaceutics-13-01415-f004:**
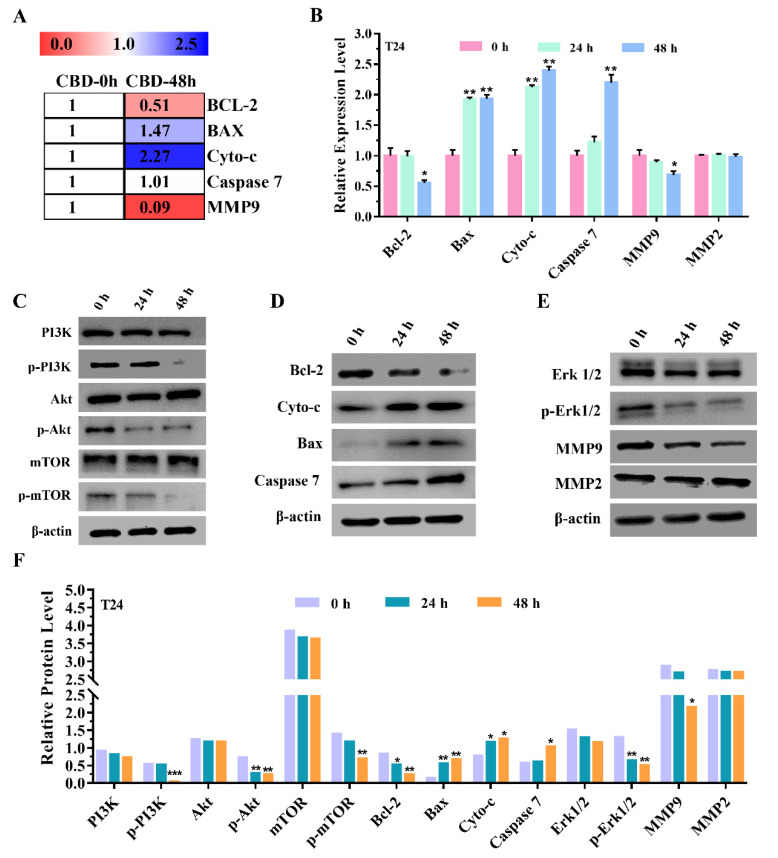
Effect of CBD on the mRNA and protein levels of cell proliferation, migration and apoptosis-related DEGs. (**A**,**B**) T24 cells were incubated with 12 µM CBD and then qRT-PCR was performed. The mRNA levels of *Bcl-2*, *Bax*, *Cyto-c*, *Caspase 7*, *MMP9*, and *MMP2* were analyzed upon exposure to CBD in T24 cells for 0, 24, and 48 h. (**C**–**E**) After 0, 24, and 48 h of treatment with CBD, the protein expression levels of PI3K, p-PI3K, Akt, p-Akt, mTOR, p-mTOR, Bcl-2, Bax, Cyto-c, Caspase 7, Erk1/2, p-Erk1/2, MMP9 and MMP2 in T24 cells were determined by Western blotting. (**F**) The ratio of protein levels was normalized according to the values of the control. All data are expressed as mean ± SD. * *p* < 0.05, ** *p* < 0.01, *** *p* < 0.001 compared to the control.

**Figure 5 pharmaceutics-13-01415-f005:**
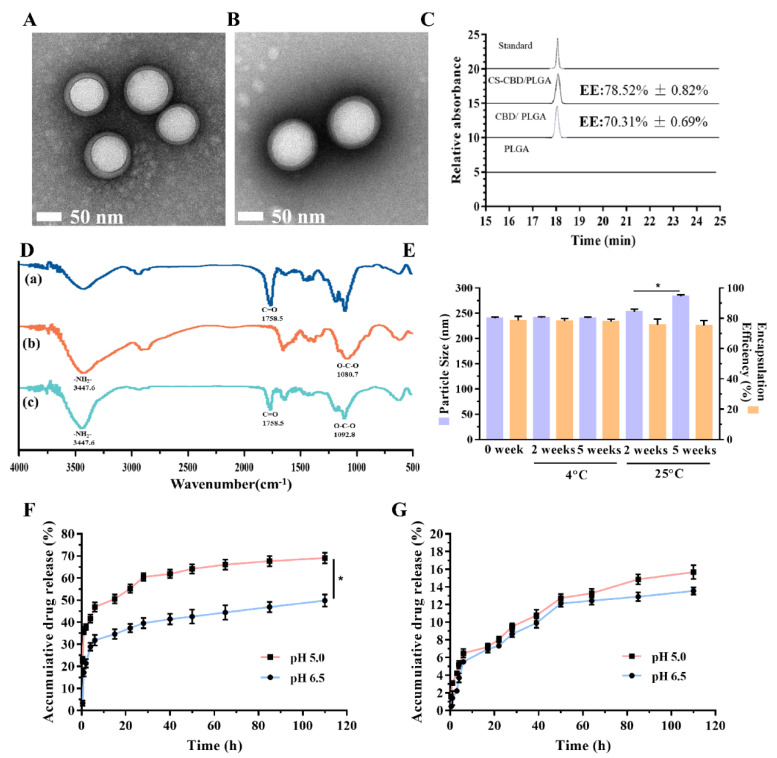
Physicochemical properties of the prepared PLGA NPs formulations. (**A**,**B**) TEM images of CBD-loaded PLGA NPs and CS-coated CBD-loaded PLGA NPs, respectively. (**C**) Encapsulation efficiency of CBD/PLGA NPs and CS-CBD/PLGA NPs were measured by HPLC. (**D**) FTIR spectra of (a) PLGA NPs, (b) CS and (c) CS-coated PLGA NPs. (**E**) Changes in particle size and EE of CS-coated CBD-loaded PLGA NPs at 4 and 25 °C for 2 and 5 weeks, respectively. In vitro cumulative release of the CBD from bare (**F**) and CS-coated (**G**) PLGA NPs in PBS for 110 h at pH 5.0 and 6.5, respectively. Data are expressed as mean ± SD (*n* = 3). * *p* < 0.05 compared to the control.

**Figure 6 pharmaceutics-13-01415-f006:**
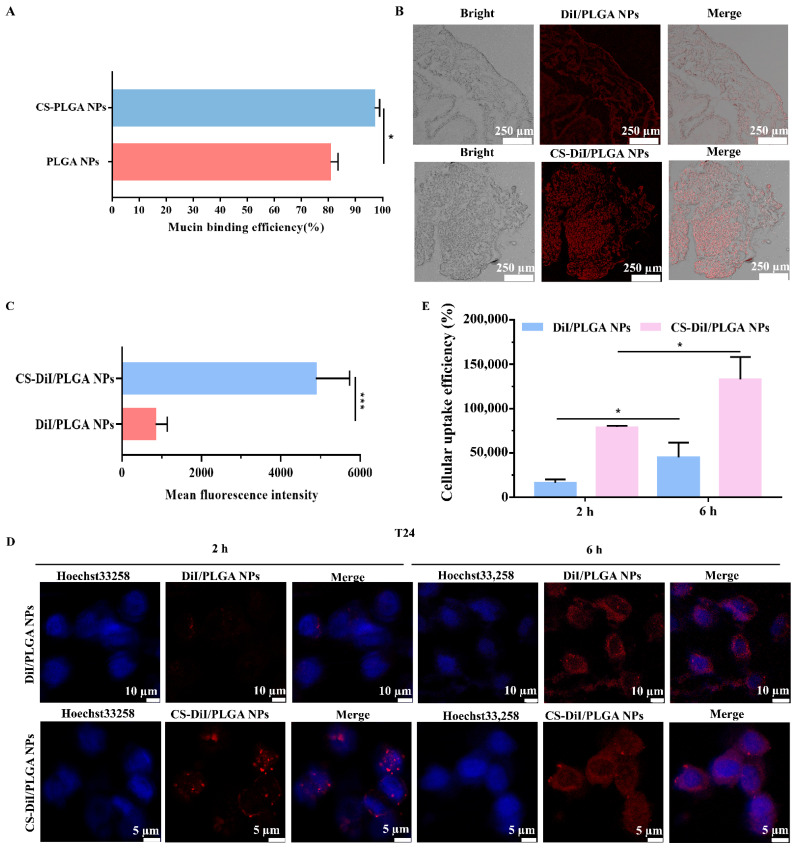
Determination of NPs adhesion and uptake ability. (**A**) Binding efficiency of mucin to NPs in vitro. Values represent the mean ± SD (*n* = 3). Statistically significant differences with PLGA NPs are marked with (*) for *p* < 0.05. (**B**) Adhesion of NPs in the mouse bladder in vivo. (**C**) The percent surface coverage by NPs on the bladder mucosa surface was quantified as the mean fluorescence intensity ± SD. (**D**) Cellular distribution of DiI/PLGA NPs and CS-DiI/PLGA NPs in T24 cells after 2 and 6 h of incubation was observed by CLSM. Confocal images of T24 cells after incubation with DiI/PLGA NPs and CS-DiI/PLGA NPs (red). Cells were labeled with Hoechst 33258 (blue) to stain the nucleus. The scale bar is 5 and 10 µm, respectively. (**E**) Cellular uptake efficiency of DiI/PLGA NPs and CS-DiI/PLGA NPs by T24 cells. Data represent the mean ± SD (*n* = 3). * *p* < 0.05, *** *p* < 0.001 compared to the control.

**Figure 7 pharmaceutics-13-01415-f007:**
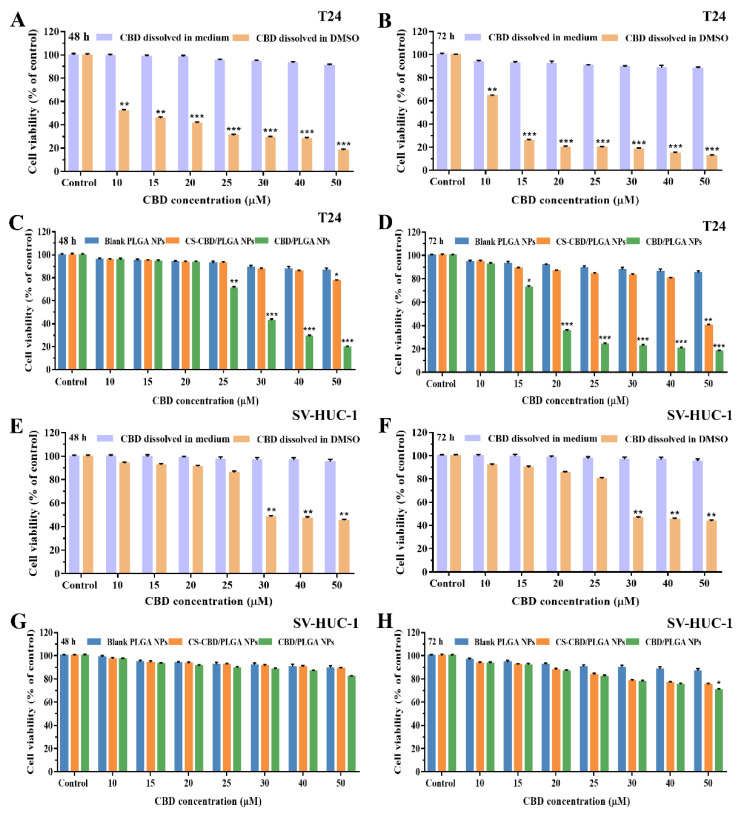
Cytotoxicity assay of bare and CS-coated CBD/PLGA NPs against T24 and SV-HUC-1 cells. The viabilities of T24 (**A**,**B)** and SV-HUC-1 (**E**,**F**) cells were measured by incubating with CBD dissolved in medium and CBD dissolved in DMSO for 48 and 72 h. The viabilities of T24 (**C**,**D**) and SV-HUC-1 (**G**,**H**) cells were measured by incubating with Blank PLGA NPs, CBD/PLGA NPs and CS-CBD/PLGA NPs for 48 and 72 h. Data are expressed as mean ± SD (*n* = 3). * *p* < 0.05, ** *p* < 0.01, and *** *p* < 0.001 compared to the control.

## Data Availability

All data needed to evaluate the conclusions in the paper are present in the paper and/or the Appendix A. Additional data related to this paper may be requested from the authors.

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
