# Peer review of "Cannabidiol Effectively Promoted Cell Death in Bladder Cancer and the Improved Intravesical Adhesion Drugs Delivery Strategy Could Be Better Used for Treatment"

_pharmaceutics, 2021, doi:10.3390/pharmaceutics13091415_

Round 1

Reviewer 1 Report

The manuscript shows intersting results and the experimental paradigms have been conducted rigorously.

However, some minor corrections have to be done before acceptance.

-Authors should better explain the KEGG analysis method in the experimental section.

-In figure captions, the ANOVA P value has to be indicated alongside with P values related to post hoc test. The use of a post hoc test has not been specified in the "Statistical analysis" paragraph. However, different P values in the caption of figure 7 reporting the results of the effects of different concentrations of CBD suggest the use of a post hoc test for unravelling concentration-dependent effects.

Authors showed that CBD stimulated early apoptosis since 12 µM concentration, in cancer cells. However, at lower concentrations (about ten-fold lower) the antiapoptotic effect was demonstrated in astrocyte cells (di Giacomo et al., 2020). Authors should consider this different response of cells to CBD, as regards its effects on early apotposis.

The references should be formatted according to journal guidelines.

Author Response

Thank you very much for regarding to our manuscript (pharmaceutics-1332169) entitled "Cannabidiol Effectively Promoted Cell Death in Bladder Cancer and the Improved Intravesical Adhesion Drugs Delivery Strategy Could be Better Used for Treatment" which was submitted to Pharmaceutics (ISSN 1999-4923).

We highly appreciate yours and the reviewers’ comments and constructive suggestions.

We have made all the possible revision according to the reviewers’ comments and suggestions that are listed as follows.

Reviewer 1:

The manuscript shows intersting results and the experimental paradigms have been conducted rigorously. However, some minor corrections have to be done before acceptance.

  1. Authors should better explain the KEGG analysis method in the experimental section.

Response to comments: Thank you for your suggestion. We have added “To identify differential expression genes (DEGs) between two different samples, the expression level of each transcript was calculated according to the transcripts per million reads (TPM) method. In addition, Kyoto Encyclopedia of Genes and Genomes (KEGG) functional-enrichment analysis was performed to identify which DEGs were significantly enriched in metabolic pathways at Bonferroni-corrected P-value ≤ 0.05 compared with the whole-transcriptome background. KEGG pathway analysis was carried out by Goatools (https://github.com/tanghaibao/Goatools) and KOBAS (http://kobas.cbi.pku.edu.cn/home.do)” in the “RNA Sequencing (RNA-seq) Assays” paragraph to explain the KEGG analysis method carefully in the revised manuscript.

  1. In figure captions, the ANOVA P value has to be indicated alongside with P values related to post hoc test. The use of a post hoc test has not been specified in the "Statistical analysis" paragraph. However, different P values in the caption of figure 7 reporting the results of the effects of different concentrations of CBD suggest the use of a post hoc test for unravelling concentration-dependent effects.

Response to comments: Thank you for your comments. As you said, the ANOVA P value has to be indicated alongside with P values related to post hoc test. We have carefully checked all date, and the statistical significance values were carried out by one-way analysis of variance (ANOVA) test with post hoc contrasts by Student-Newman-Keuls test again. Meanwhile, we have revised the “Statistical analysis” paragraph in the original manuscript.

  1. Authors showed that CBD stimulated early apoptosis since 12 µM concentration, in cancer cells. However, at lower concentrations (about ten-fold lower) the antiapoptotic effect was demonstrated in astrocyte cells (di Giacomo et al., 2020). Authors should consider this different response of cells to CBD, as regards its effects on early apotposis.

Response to comments: Thank you for your comments. Giacomo et al. (2020, PMID: 32443623) reported that CBD exerted an antiapoptotic effect in CTX-TNA2 astrocyte cell line at a concentration ranging from 1 nM to 1000 nM. However, in the present study, we found CBD stimulated early apoptosis in bladder cancer cells at 12 µM concentration. In fact, there have been several studies showed that CBD promoted early apoptosis. For example, Elbaz et al. (2015, PMID: 25660577) indicated that CBD inhibited proliferation of breast cancer cells at 9 µM, 12 µM, and 15 µM, significantly. Sultan et al. (2018, PMID: 30007266) showed that the IC50 of CBD for MDA-MB-231 and T-47D cells were 2.2 µM and 5 µM, and induce cell apoptosis at this concentration. We are temporarily unable to explain that low concentrations of CBD have antiapoptosis effect in astrocyte cells, while high concentrations promote apoptosis in several cancer cells, but we speculate that there may be several reasons. Firstly, different cell lines lead to different biological responses to CBD. Secondly, there may be some anti-apoptotic mechanism in cells treated with low concentration of CBD. Thirdly, the chemical structure of CBD. For instance, Campos et al. (2016, PMID: 26845349) proposed that CBD could show neuroprotective effects by decreasing oxidative parameters and increasing cell viability, according to its chemical structure.  As a result, your constructive comments provide us with new research directions. In the further study, we will conduct in-depth research on whether the concentration of CBD in different cells promotes or inhibits early apoptosis.

  1. The references should be formatted according to journal guidelines.

Response to comments: Thank you for your reminder. We have updated and checked the references format according to journal guidelines.

All the comments and suggestions have been considered in the revised version.

I do hope this revised manuscript will make you feel more satisfactory.

If you have any further questions regarding the revised version, please do not hesitate to contact me. I am looking forward to hearing from you soon.

Thank you very much and with my best regards.

Yours sincerely,

Xingyuan Ma(corresponding author)

Professor of Biomedicine, School of Biological Engineering

State Key Laboratory of Bioreactor Engineering

East China University of Science and Technology

130 Meilong Road, P.O. Box No. 365, Shanghai, 200237, P. R. China

Tel: 86-21-64253980

Email: maxy@ecust.edu.cn

Reviewer 2 Report

Congratulations on a very interesting work. The manuscript is very well developed, the results are correctly and interestingly described. There are many results, which only adds value to the work. The potential use of cannabidiol is extremely important from the point of view of the development of potential anticancer therapeutics. Cannabidiol is a new therapeutic challenge, especially in terms of its wide possibilities.

The work is original, it brings new knowledge to science, a consistently conducted and appropriately selected methodology has been used, it does not contain unverified research results, and the given literature is the latest relevant publications recognized in a given field.

There is only one shortcoming in my opinion. The presented figures are very difficult to read. I believe that they should be enlarged, which would improve the quality of work.

Author Response

Thank you very much for regarding to our manuscript (pharmaceutics-1332169) entitled "Cannabidiol Effectively Promoted Cell Death in Bladder Cancer and the Improved Intravesical Adhesion Drugs Delivery Strategy Could be Better Used for Treatment" which was submitted to Pharmaceutics (ISSN 1999-4923).

We highly appreciate yours and the reviewers’ comments and constructive suggestions.

We have made all the possible revision according to the reviewers’ comments and suggestions that are listed as follows.

Reviewer 2:

Congratulations on a very interesting work. The manuscript is very well developed, the results are correctly and interestingly described. There are many results, which only adds value to the work. The potential use of cannabidiol is extremely important from the point of view of the development of potential anticancer therapeutics. Cannabidiol is a new therapeutic challenge, especially in terms of its wide possibilities.

The work is original, it brings new knowledge to science, a consistently conducted and appropriately selected methodology has been used, it does not contain unverified research results, and the given literature is the latest relevant publications recognized in a given field.

There is only one shortcoming in my opinion. The presented figures are very difficult to read. I believe that they should be enlarged, which would improve the quality of work.

Response to comments: Thank you for your affirmation. We have enlarged part of the figures according to your requirements, and we have uploaded the original high-definition figures in a compressed package.

All the comments and suggestions have been considered in the revised version.

I do hope this revised manuscript will make you feel more satisfactory.

If you have any further questions regarding the revised version, please do not hesitate to contact me. I am looking forward to hearing from you soon.

Thank you very much and with my best regards.

Yours sincerely,

Xingyuan Ma(corresponding author)

Professor of Biomedicine, School of Biological Engineering

State Key Laboratory of Bioreactor Engineering

East China University of Science and Technology

130 Meilong Road, P.O. Box No. 365, Shanghai, 200237, P. R. China

Tel: 86-21-64253980

Email: maxy@ecust.edu.cn

Reviewer 3 Report

This is a widespread study on effect on cannabinoids on various bladder cancer cell lines. Anticancer properties are studied. A number of laboratory techniques are used.
1.Microphotographic documentation and graphs are too small. Some microphotographs are of different magnifications and bars are the same , please check
2.Provide passage number, and information on assays you use in your laboratory to check whether the cell lines maintain their molecular and morphological properties
3. Provide biological characteristics (morphological and biochemical properties) of cell lines , are their transformed, tumor etc?
4. Provide concentrations of antibodies used for analyses
4. It will be nice to have a compact conclusion of the study included

Author Response

Thank you very much for regarding to our manuscript (pharmaceutics-1332169) entitled "Cannabidiol Effectively Promoted Cell Death in Bladder Cancer and the Improved Intravesical Adhesion Drugs Delivery Strategy Could be Better Used for Treatment" which was submitted to Pharmaceutics (ISSN 1999-4923).

We highly appreciate yours and the reviewers’ comments and constructive suggestions.

We have made all the possible revision according to the reviewers’ comments and suggestions that are listed as follows.

Reviewer 3:

This is a widespread study on effect on cannabinoids on various bladder cancer cell lines. Anticancer properties are studied. A number of laboratory techniques are used.

  1. Microphotographic documentation and graphs are too small. Some microphotographs are of different magnifications and bars are the same , please check.

Response to comments: Thank you for your comments. We have made appropriate adjustments to the documents and graphs. Meanwhile, we have double-checked the bars again. The length of the bar in the enlarged figure is actually different. To be more intuitive, we have bolded the bar and marked the corresponding value on the left of the bar in the revised manuscript.

  1. Provide passage number, and information on assays you use in your laboratory to check whether the cell lines maintain their molecular and morphological properties.

Response to comments: Thank you for your comments. All bladder cell lines were supported by Sun Yat-Sen memorial hospital (Sun Yat-Sen university). The cells given to us have been passed through for 18 generations, and the cells we did experiments on were all 20 generations. Meanwhile, Sun Yat-sen University also provided us with the STR typing test report of T24 (provided by Guangzhou IGE Biotechnology Ltd), UM-UC-3 (provided by Guangzhou IGE Biotechnology Co., Ltd), 5637 (YuChiCell (Shanghai) Biological Technology Co., Ltd), and SV-HUC-1 cell lines (Shanghai Zhong Qiao Xin Zhou Biotechnology Co., Ltd.) (attached below). And we have always maintained a rigorous attitude to ensure that there is no cross-contamination phenomenon in all experiments.

T24:

UM-UC-3:

5637:

SV-HUC-1:

  1. Provide biological characteristics (morphological and biochemical properties) of cell lines , are their transformed, tumor etc?

Response to comments: Thank you for your comments. We have summarized the results into a table as shown below. And we have put the following table in the supplementary data.

Cell

T24

UM-UC-3

5637

SV-HUC-1

Organism

Homo sapiens

Morphology

epithelial

Tissue

Urinary bladder

Ureter; Uroepithelium

Growth properties

Adherent

Gender

Female

Male

Disease

Transitional Cell Carcinoma

Grade II Carcinoma

No

Tumorigenic

Yes, in hamster cheek pouch;

No, in nude mice

Yes, Tumors developed within 21 days at 100% frequency (5/5) in nude mice inoculated subcutaneously with 107 cells.

Yes, within 21 days at 100% frequency (5/5) in nude mice inoculated subcutaneously with 107 cells.

 No, nude mice

  1. Provide concentrations of antibodies used for analyses.

Response to comments: Thank you for your comments. We have provided the concentrations of antibodies in the revised manuscript. For example, Antibodies against PI3K (1500 μg/mL, 1: 10000), Akt (2504 μg/mL, 1: 3000), phospho-Akt (1683 μg/mL, 1: 6000), mTOR (500 μg/mL, 1: 10000), phospho-mTOR (1000 μg/mL, 1: 1000), Bcl-2 (1000 μg/mL, 1: 2000), Bax (1600 μg/mL, 1: 4000), Cytochrome-c (1000 μg/mL, 1: 4000), Caspase 7 (700 μg/mL, 1: 1000), Erk1/2 (1000 μg/mL, 1: 6000), phospho-Erk1/2 (370 μg/mL, 1: 3000), MMP9 (400 μg/mL, 1: 600), MMP2 (1000 μg/mL, 1: 600), and β-actin (427 μg/mL, 1: 5000) were purchased from Proteintech (Shanghai, China). Also, phos-pho-PI3K (690 μg/mL, 1: 1000) was purchased from Abcolonal (Shanghai, China).

  1. It will be nice to have a compact conclusion of the study included.

Response to comments: Thank you for your nice suggestion. We have added “In conclusion, MTT and colony forming assays showed that CBD could inhibit proliferation on BC cells, including T24, UM-UC-3, and 5637, especially at 12 µM, the effect on T24 was the most significant. Moreover, cell apoptosis was detected by using nuclear staining, TUNEL, and flow cytometer. Subsequently, wound healing assay suggested that CBD reduced the migration ability of BC cells. In order to explore the potential reasons of CBD to cause the results above, the CBD-treated T24 cells transcriptome analysis was conducted. RNA sequencing results indicated that several genes related to apoptosis, proliferation, and migration have undergone significant changes, such as Bax, Cyto-c, Bcl-2, and Caspase 7. Further analysis by KEGG found that the significantly changed genes were enriched in the PI3K/Akt signaling pathway. Therefore, these related genes were screened from the transcriptome databases and confirmed their expression levels by qRT-PCR and western blotting. These results indicated that CBD increased Bax, Cyto-c, and Caspase 7 levels, whereas decreased Bcl-2, p-PI3K, p-Akt, p-mTOR, p-Erk1/2 and MMP9. Taken together, CBD could inhibit BC cells growth, mi-gration, and induce apoptosis by inactivating of the PI3K/Akt pathway. Meanwhile, using chitosan to wrap CBD-loaded PLGA nanoparticles could significantly enhance the adhesion of CBD in the bladder wall, which might not only avoid damage caused by repeated perfusion of organic solvents, but also achieve the purpose of longterm treatment. We believe that our study makes a significant contribution to the field because these results could be developed as a promising strategy for a safer and more efficient anticancer therapy” as a compact conclusion of the study in the revised manuscript.

All the comments and suggestions have been considered in the revised version.

I do hope this revised manuscript will make you feel more satisfactory.

If you have any further questions regarding the revised version, please do not hesitate to contact me. I am looking forward to hearing from you soon.

Thank you very much and with my best regards.

Yours sincerely,

Xingyuan Ma(corresponding author)

Professor of Biomedicine, School of Biological Engineering

State Key Laboratory of Bioreactor Engineering

East China University of Science and Technology

130 Meilong Road, P.O. Box No. 365, Shanghai, 200237, P. R. China

Tel: 86-21-64253980

Email: maxy@ecust.edu.cn

Round 2

Reviewer 3 Report

My comments were considered however I still think that micro photographic documentation should be larger

Author Response

Reviewer 3:

My comments were considered however I still think that micro photographic documentation should be larger.

Response to comments: We appreciate for your suggestion and comments. Regarding the size of the micro photographic documentation, we have enlarged it according to your requirement in the revised manuscript.
